# Association of Selenoprotein and Selenium Pathway Genotypes with Risk of Colorectal Cancer and Interaction with Selenium Status

**DOI:** 10.3390/nu11040935

**Published:** 2019-04-25

**Authors:** Veronika Fedirko, Mazda Jenab, Catherine Méplan, Jeb S. Jones, Wanzhe Zhu, Lutz Schomburg, Afshan Siddiq, Sandra Hybsier, Kim Overvad, Anne Tjønneland, Hanane Omichessan, Vittorio Perduca, Marie-Christine Boutron-Ruault, Tilman Kühn, Verena Katzke, Krasimira Aleksandrova, Antonia Trichopoulou, Anna Karakatsani, Anastasia Kotanidou, Rosario Tumino, Salvatore Panico, Giovanna Masala, Claudia Agnoli, Alessio Naccarati, Bas Bueno-de-Mesquita, Roel C.H. Vermeulen, Elisabete Weiderpass, Guri Skeie, Therese Haugdahl Nøst, Leila Lujan-Barroso, J. Ramón Quirós, José María Huerta, Miguel Rodríguez-Barranco, Aurelio Barricarte, Björn Gylling, Sophia Harlid, Kathryn E. Bradbury, Nick Wareham, Kay-Tee Khaw, Marc Gunter, Neil Murphy, Heinz Freisling, Kostas Tsilidis, Dagfinn Aune, Elio Riboli, John E. Hesketh, David J. Hughes

**Affiliations:** 1Department of Epidemiology, Rollins School of Public Health & Winship Cancer Institute, Emory University, Atlanta, GA 30322, USA; veronika.fedirko@emory.edu (V.F.); jeb.jones@emory.edu (J.S.J.); WZHU4@emory.edu (W.Z.); 2Section of Nutrition and Metabolism, International Agency for Research on Cancer, 69372 Lyon, France; jenabm@iarc.fr (M.J.); GunterM@iarc.fr (M.G.); MurphyN@iarc.fr (N.M.); FreislingH@iarc.fr (H.F.); 3School of Biomedical Sciences, Newcastle University, Newcastle upon Tyne NE1 7RU, UK; Catherine.Meplan@newcastle.ac.uk (C.M.); j.hesketh@rgu.ac.uk (J.E.H.); 4Institute for Experimental Endocrinology, University Medical School, D-13353 Berlin, Germany; lutz.schomburg@charite.de (L.S.); sandra.hybsier@charite.de (S.H.); 5Department of Epidemiology and Biostatistics, The School of Public Health, Imperial College London, London W2 1PG, UK; afshan.siddiq@genomicsengland.co.uk (A.S.); bas.bueno.de.mesquita@rivm.nl (B.B.-d.-M.); ktsilidis@gmail.com (K.T.); d.aune@imperial.ac.uk (D.A.); e.riboli@imperial.ac.uk (E.R.); 6Department of Public Health, Section for Epidemiology, Aarhus University, 8000 Aarhus, Denmark; ko@dce.au.dk; 7Diet, Genes and Environment Unit, Danish Cancer Society Research Center, DK 2100 Copenhagen, Denmark; annet@cancer.dk; 8Faculty of Medicine, CESP, University of Paris-Sud, Faculty of Medicine UVSQ, INSERM, University of Paris-Saclay, 94805 Villejuif, France; HANANE.OMICHESSAN@gustaveroussy.fr (H.O.); vittorio.perduca@gmail.com (V.P.); boutron@igr.fr (M.-C.B.-R.); 9Centre for Research in Epidemiology and Population Health (CESP), F-94805 Gustave Roussy, Villejuif, France; 10Laboratory of Applied Mathematics, MAP5 (UMR CNRS 8145), University of Paris Descartes, 75270 Paris, France; 11Division of Cancer Epidemiology, German Cancer Research Centre (DKFZ), 69120 Heidelberg, Germany; t.kuehn@dkfz-Heidelberg.de (T.K.); v.katzke@dkfz-Heidelberg.de (V.K.); 12Department of Epidemiology, German Institute of Human Nutrition Potsdam-Rehbrücke, 14558 Nuthetal, Germany; Krasimira.Aleksandrova@dife.de; 13Hellenic Health Foundation, 115 27 Athens, Greece; atrichopoulou@hhf-greece.gr (A.T.); a.karakatsani@hhf-greece.gr (A.K.); a.kotanidou@hhf-greece.gr (A.K.); 142nd Pulmonary Medicine Department, School of Medicine, National and Kapodistrian University of Athens, “ATTIKON” University Hospital, 106 79 Haidari, Greece; 151st Department of Critical Care Medicine and Pulmonary Services, University of Athens Medical School, Evangelismos Hospital, 106 76 Athens, Greece; 16Cancer Registry and Histopathology Department, Civic M.P. Arezzo Hospital, 97100 Ragusa, Italy; rtumino@tin.it; 17Department of Clinical Medicine and Surgery, Federico II University, 80138 Naples, Italy; spanico@unina.it; 18Cancer Risk Factors and Life-Style Epidemiology Unit, Cancer Research and Prevention Institute—ISPO, 50141 Florence, Italy; g.masala@ispo.toscana.it; 19Epidemiology and Prevention Unit, IRCCS Foundation National Cancer Institute, 20133 Milan, Italy; claudia.agnoli@istitutotumori.mi.it; 20Molecular and Genetic Epidemiology Unit, Italian Institute for Genomic Medicine (IIGM) Torino, 10126 Torino, Italy; alessio.naccarati@hugef-torino.org; 21Department for Determinants of Chronic Diseases (DCD), National Institute for Public Health and the Environment (RIVM), 3720 Bilthoven, The Netherlands; 22Department of Gastroenterology and Hepatology, University Medical Centre, 3584 CX Utrecht, The Netherlands; 23Department of Social and Preventive Medicine, Faculty of Medicine, University of Malaya, Kuala Lumpur 50603, Malaysia; 24Institute of Risk Assessment Sciences, Utrecht University, 3512 JE Utrecht, The Netherlands; R.C.H.Vermeulen@uu.nl; 25Department of Research, Cancer Registry of Norway, Institute of Population-Based Cancer Research, N-0304 Oslo, Norway; WeiderpassE@iarc.fr; 26Department of Medical Epidemiology and Biostatistics, Karolinska Institute, SE-171 77 Stockholm, Sweden; 27Genetic Epidemiology Group, Folkhälsan Research Center, and Faculty of Medicine, Helsinki University, 00014 Helsinki, Finland; 28Department of Community Medicine, University of Tromsø, The Arctic University of Norway, 9019 Tromsø, Norway; Guri.Skeie@ism.uit.no (G.S.); therese.h.nost@uit.no (T.H.N.); 29Unit of Nutrition and Cancer, Catalan Institute of Oncology (ICO-IDIBELL), L’Hospitalet de Llobregat, 08908 Barcelona, Spain; llujan@iconcologia.net; 30EPIC Asturias, Public Health Directorate, 33006 Oviedo, Asturias, Spain; joseramon.quirosgarcia@asturias.org; 31Department of Epidemiology, Murcia Regional Health Council, IMIB-Arrixaca, 30008 Murcia, Spain; jmhuerta.carm@gmail.com; 32CIBER Epidemiology and Public Health (CIBERESP), 28029 Madrid, Spain; miguel.rodriguez.barranco.easp@juntadeandalucia.es (M.R.-B.); aurelio.barricarte.gurrea@cfnavarra.es (A.B.); 33Andalucia School of Public Health, Institute for Biosanitary Research, University Hospital of Granada, University of Granada, 18011 Granada, Spain; 34Epidemiology, Prevention and Promotion Health Service, Navarra Public Health Institute, 31003 Pamplona, Spain; 35Navarra Institute for Health Research (IdiSNA), 31008 Pamplona, Spain; 36Department of Medical Biosciences, Pathology, Umea University, 901 87 Umea, Sweden; bjorn.gylling@umu.se; 37Department of Radiation Sciences, Oncology, Umea University, 901 87 Umea, Sweden; sophia.harlid@umu.se; 38Cancer Epidemiology Unit, Nuffield Department of Population Health, University of Oxford, Oxford OX3 7LF, UK; kathryn.bradbury@ceu.ox.ac.uk; 39MRC Epidemiology Unit, University of Cambridge, CB2 0QQ Cambridge, UK; nick.wareham@mrc-epid.cam.ac.uk; 40School of Clinical Medicine, University of Cambridge, Clinical Gerontology Unit, Addenbrooke’s Hospital, Cambridge CB2 0QQ, UK; kk101@medschl.cam.ac.uk; 41Department of Hygiene and Epidemiology, University of Ioannina School of Medicine, 45110 Ioannina, Greece; 42Department of Nutrition, Bjørknes University College, 0456 Oslo, Norway; 43Department of Endocrinology, Morbid Obesity and Preventive Medicine, Oslo University Hospital, 0372 Oslo, Norway; 44Cancer Biology and Therapeutics Group, UCD Conway Institute, School of Biomolecular and Biomedical Science, University College Dublin, D04 V1W8 Dublin, Ireland

**Keywords:** selenium, selenium status, selenoprotein gene variation, selenium pathway, colorectal neoplasms, selenoprotein P, prospective cohort, colorectal cancer risk, genetic epidemiology, biomarkers

## Abstract

Selenoprotein genetic variations and suboptimal selenium (Se) levels may contribute to the risk of colorectal cancer (CRC) development. We examined the association between CRC risk and genotype for single nucleotide polymorphisms (SNPs) in selenoprotein and Se metabolic pathway genes. *Illumina Goldengate* assays were designed and resulted in the genotyping of 1040 variants in 154 genes from 1420 cases and 1421 controls within the European Prospective Investigation into Cancer and Nutrition (EPIC) study. Multivariable logistic regression revealed an association of 144 individual SNPs from 63 Se pathway genes with CRC risk. However, regarding the selenoprotein genes, only *TXNRD1* rs11111979 retained borderline statistical significance after adjustment for correlated tests (*P*_ACT_ = 0.10; *P*_ACT_ significance threshold was *P* < 0.1). SNPs in Wingless/Integrated (Wnt) and Transforming growth factor (TGF) beta-signaling genes (*FRZB*, *SMAD3*, *SMAD7*) from pathways affected by Se intake were also associated with CRC risk after multiple testing adjustments. Interactions with Se status (using existing serum Se and Selenoprotein P data) were tested at the SNP, gene, and pathway levels. Pathway analyses using the modified Adaptive Rank Truncated Product method suggested that genes and gene x Se status interactions in antioxidant, apoptosis, and TGF-beta signaling pathways may be associated with CRC risk. This study suggests that SNPs in the Se pathway alone or in combination with suboptimal Se status may contribute to CRC development.

## 1. Introduction

In Europe, colorectal cancer (CRC) is the cancer type with both the second highest incidence and mortality rate [1]. Substantial CRC risk may derive from dietary factors, genetic variants, and their interactions [2,3].

Experimental and observational evidence suggests that suboptimal dietary intakes of the micronutrient selenium (Se) contribute to greater risk for the development of cancers at several anatomical sites, including the colorectum [4,5,6]. In humans, Se exerts its potential anti-carcinogenic properties through incorporation into 25 selenoproteins by the amino acid selenocysteine [7,8]. Several selenoproteins protect cells from damaging oxidative radicals including the glutathione peroxidases (notably GPX1 and GPX4), components of the thioredoxin reductase system (TXNRD1-3) and selenoprotein P (SELENOP; please note the modified selenoprotein nomenclature [9]) which is also critical for Se transport [8,10,11].

The major mechanism through which Se is thought to influence the risk of CRC development is variation in gene expression and biosynthesis of protective selenoproteins [12,13]. In rodent models, adequate Se intake and selenoprotein expression have been shown to prevent colon cancer while selenoprotein dysregulation may increase colon cancer risk [14,15,16]. Data from nutritional intervention trials and epidemiological studies suggest implications for Se intake regarding CRC risk could potentially be more important in individuals with particular selenoprotein genotypes and/or in populations with low Se status, such as in Western Europe where the present study was conducted [4,5,17,18]. Risk modification by sex has also been observed for CRC risk associations with selenoprotein genotypes [19,20] and Se status [5,17].

Genetic variations in approximately half of the 25-known human selenoprotein genes have been associated with susceptibility to CRC and/or colorectal adenoma (CRA) risk in at least seven populations from Asia, North America and Europe; in addition some of these variants have been shown to impact survival outcomes (reviewed in [4,21]). Although some of these studies have been performed in suboptimal Se intake areas, large studies have more generally been conducted in Se-replete environments in North America and these have reported evidence both for [22,23] and against [17] an association of selenoprotein genes with CRC risk. However, only a limited number of single nucleotide polymorphisms (SNPs) in selected selenoprotein genes have been analyzed, while in several of these reports the Se status of the analyzed cohort was not assessed.

To our knowledge, it is unknown which selenoproteins are critical in maintaining colonic health and no study has comprehensively evaluated variation in all selenoprotein genes for association with CRC risk. Moreover, interactions of selenoprotein genetic variations according to robust Se status biomarkers have not been explored. As both genetic factors and dietary Se intake can influence the pattern of selenoprotein expression and biosynthesis, we hypothesized that variation in selenoprotein genes, and in related signaling pathway genes influenced by Se intake (together comprising the ‘Se pathway’), affect CRC development risk, while Se status may modify this risk.

In this study, we have examined for the first time the association of detailed Se pathway gene variation with cancer risk in 1420 CRC cases and 1421 controls within the European Prospective Investigation into Cancer and Nutrition (EPIC) cohort. We previously reported in a subset of this nested cohort with 966 case-control pairs that a higher Se status (ascertained by serum levels of Se and SELENOP) was associated with a lower CRC risk [5]. In these Western European subjects, the mean Se and SELENOP circulating levels were 84.0 μg/L and 4.3 mg/L in cases and 85.6 μg/L and 4.4 mg/L in controls, respectively. Thus, our present study was conducted in a generally suboptimal Se status population, as these Se concentrations are insufficient for optimal GPX3 expression and SELENOP saturation [5,6]. We now report the interaction between these genes and their corresponding pathways with Se status biomarkers and CRC risk.

## 2. Materials and Methods

### 2.1. Study Population and Design

EPIC is a multicenter prospective cohort study designed to investigate the association between diet, lifestyle, genetic and environmental factors and the incidence of cancers. The rationale and methods of the EPIC design have been described previously [24,25]. Briefly, 521,448 men and women mostly aged 25–70 years were enrolled between 1992–2000 in 23 sub-cohorts in 10 European countries (Denmark, France, Germany, Greece, Italy, The Netherlands, Norway, Spain, Sweden, and United Kingdom). The present analysis is based on participant data from all sub-cohorts except for Norway. At recruitment, standardized dietary, lifestyle and socio-demographic questionnaires including information on physical activity, education, smoking and medical history; anthropometric data, and blood samples were collected from participants. Blood and DNA samples are stored at the International Agency for Research on Cancer (IARC, Lyon, France) at −196 °C under liquid nitrogen for all countries except Denmark (−150 °C, nitrogen vapor) and Sweden (−80 °C freezers). Sample storage standardization including DNA extraction and quantification protocols were previously described in [26].

All study participants provided written informed consent. Ethical approval for the EPIC study was obtained from the review boards of the IARC (IARC Ethics Committee) and local participating centers. Study design methods were performed in accordance with the STROBE (Strengthening the Reporting of Observational Studies in Epidemiology) guidelines (https://www.strobe-statement.org/index.php?id=strobe-home).

### 2.2. Follow-Up for Cancer Incidence

Cancer incidence was determined through record linkage with population-based cancer registries (Denmark, Italy, Netherlands, Spain, Sweden, United Kingdom) or via a combination of methods, including the use of health insurance records, cancer and pathology registries, and active contact of study subjects or next-of-kin (France, Germany, Greece). Complete follow-up censoring dates for this study varied among centers, ranging between June 2002 and June 2003.

### 2.3. Selection of Cases and Controls and Study Design

Case subjects were men and women who developed first incident CRC after recruitment and before the latest follow-up date. Cancer incidence data were coded using the 10th Revision of the International Classification of Diseases (ICD-10) and the second revision of the International Classification of Disease for Oncology (ICDO-2). Colon cancers were defined as tumors in the cecum, appendix, ascending colon, hepatic flexure, transverse colon, splenic flexure, descending and sigmoid colon (C18.0-C18.7), and overlapping or unspecified origin tumors (C18.8 and C18.9). Rectal cancers were defined as tumors occurring at the recto-sigmoid junction (C19) or rectum (C20). Anal canal cancers (C21) were excluded. Colorectal cancer is the combination of the colon and rectal cancer cases.

All subjects with prior cancer diagnosis at any site (except non-melanoma skin cancer) were excluded. Cases were matched 1:1 by study center of enrollment, sex, age at blood collection, time of blood collection and fasting status, and menopausal status among women. Premenopausal women were matched on phase of menstrual cycle and postmenopausal women were matched on current hormonal therapy (HT) use. The matching was done as part of a previously published study on Se status [5], except for cases from Denmark for which new control subjects were identified due to problems with accessing the biobank. Furthermore, additional newly identified cases with their matching controls were also included for the genotyping from all participating countries but did not have biomarkers of Se status. Sweden was the only country of the nine participating in the genetic analysis for which we had no Se status data. Hence, there were 1478 cases and 1478 controls available for genotyping but the Se status information was only available for 966 of the cases and for 966 of the controls.

### 2.4. Gene Selection and Rationale

To examine selenoprotein gene and wider Se pathway gene variations in relation to CRC risk, we selected 1264 functional and haplotype tagging SNPs (tagSNPs) to comprehensively analyze common SNP variation in 164 Se pathway genes, which we assigned into eight functional pathways (listed in Appendix A). These included 42 genes in the primary selenoprotein pathway 1 (25 selenoprotein genes and 17 genes involved in Se transport and metabolism), and 122 genes in pathways 2–8 from (i) pathways affected by Se intake (Wnt, mTOR, Nrf2 and NF-κB signaling, endoplasmic reticulum and oxidative stress responses), and (ii) associated pathways of inflammatory response, apoptosis, DNA repair, Transforming growth factor (TGF) beta-signaling, and cell-cycle control [12] as detailed in Méplan and Hesketh, 2012 [13]. Variants in several genes from these affiliated pathways have been associated with CRC risk including regions of the Wingless/Integrated (Wnt) signaling gene *C-MYC* in CRC genome-wide association studies (GWAS) [27]. Our SNP analysis set was substantially enlarged from and included the 384 Se pathway SNPs (in 72 Se related genes) Se ‘SNP-Chip’ devised for a similar study of gene-Se interaction in a prostate cancer study within EPIC [28].

### 2.5. Tagging Single Nucleotide Polymorphism (tagSNP) Selection Protocol

A list of SNPs in all gene regions was compiled using the data from HapMap (release 27, based on dbSNP version b126 and NCBI genome build 36). TagSNPs were selected by use of the Tagger algorithm as implemented in the Haploview 3.2 software (Broad Institute, Cambridge, MA, USA). Parameters used for SNP selection were a Minor Allele Frequency (MAF) ≥5% in Caucasians and pairwise tagging (r^2^ ≥ 0.8). To include SNPs in promoter and potential regulatory regions, +/− 2 to 5 kilo base-pairs beyond the 5′ and 3′ ends were included. Additionally, known functional variants in our selected genes were added to the tagSNP list, e.g., for the selenoproteins these included rs7579, rs297299, and rs3877899 in *SELENOP* [4]. Selected SNPs were then assessed for suitability for the *Illumina GoldenGate^TM^* (Saffron Walden, Essex, UK) genotyping platform using Illumina’s custom assay building platform (https://www.illumina.com/Documents/products/technotes/technote_goldengate_design.pdf). Fifty-five SNPs which failed assay development criteria were replaced by proxy SNPs, i.e., those within the same genic region in high LD (r^2^ > 0.8) to the original SNP. Proxy SNP replacements for functional selenoprotein SNPs which failed assay design included rs1800668 for *GPX1*-rs1050450, and rs5845 plus rs540049 for *SELENOF*-rs5859. However, there were no adequate proxies for *SELENOS*-rs34713741 or *GPX4*-rs713041.

### 2.6. Genotyping

A total of 1264 SNPs from 164 Se pathway genes were genotyped by *Illumina Goldengate^TM^* in DNA samples available for 1478 case-control pairs matched within EPIC. Genotyping was performed simultaneously for cases and controls, blinded to case-control status (but with matched pairs analyzed in the same batch). A total of 62 replicate samples were genotyped to test for internal quality control, approximately 2 per genotyping plate, with the lowest reproducibility frequency for each of the replicates of 0.98. Samples with unclear or failed genotype calls were excluded from the analysis, leaving 1420 cases and 1421 controls for subsequent analyses.

From the 1264 initially selected, 96 SNPs failed genotyping, 27 failed Hardy–Weinberg Equilibrium (HWE), and 101 had less than 80% successfully genotyped samples. Thus, 1040 SNPs in 154 Genes (24 selenoprotein genes analyzed of 25, and 130 other Se pathways genes) with at least 80% genotypes across all genotyped samples were included in the final dataset (with a final genotyping call rate of 0.97, excluding zero call rate and those removed). Appendix A provides the full gene and SNP list successfully analyzed in the current study.

### 2.7. Selenium Status Assays

Measurements of serum Se and SELENOP were previously done for a subset (966 cases and 966 controls) of the current analyzed cohort. The methods used were described in Hughes et al., 2015 and Hybsier et al., 2017 [5,29]. Briefly, total Se levels were measured in 4 uL of each serum sample using a bench-top total reflection X-ray fluorescence (TXRF) spectrometer (Picofox^TM^ S2, Bruker Nano GmbH, Berlin, Germany). SELENOP protein concentrations were ascertained from 20 μL of each serum sample by a colorimetric enzyme-linked immunoassay (Selenotest™, ICI GmbH, Berlin, Germany). For quality-control, the sample type (case or control) was blinded and two serum samples of known Se and SELENOP concentrations for intra-assay variability were included in each analysis plate. The samples were measured in duplicate and the mean concentration values, standard deviation (SD), and coefficient of variation (CV) were calculated. Duplicate samples with variances in concentration over 10% were re-measured. The evaluation was performed using GraphPad Prism 6.01 (GraphPad Software, La Jolla, CA, USA) using a four-parameter logistic function. The CV was 7.3% and 7.2% for controls 1 (SELENOP: 1.5 mg/L) and 2 (SELENOP: 8.6 mg/L), respectively.

### 2.8. Statistical Analysis

Both unconditional and conditional logistic regression analysis were carried out to assess the association of individual SNPs with CRC risk, adjusting for age (as a continuous variable), sex, and study center and provided similar results. We present the data for the unconditional logistic regression. Four standard genetic analysis models were tested for disease penetrance: multiplicative, additive, common recessive, and common dominant models [30]. Sub-group analyses by sex and anatomical sub-site of the colorectum (colon and rectum) were conducted. The associations between Se and SELENOP concentrations and genetic variants (coded as 0, 1, 2 corresponding to the number of minor alleles) were assessed among controls using linear regression models adjusted for age, sex, and center. Further adjustment by body mass index (BMI), smoking status, and physical activity did not change the results substantially.

Multiple testing corrections were performed by the Benjamini–Hochberg (BH) procedure [31]. *P*-values were also adjusted for correlated tests (*P*_ACT_) to take account of the correlated nature of the SNP data in biologically relevant and related pathways [32]. BH was performed for all SNPs, followed by *P*_ACT_ for the genes that had SNPs with *P* < 0.01.

We further employed exploratory gene- and pathway-based testing based on overall SNP variation to help identify possible important Se related biological pathways and genes with multiple risk variants that may be discounted in multiple testing corrections for the large number of SNPs with small effect sizes in a SNP by SNP approach. Genes were classified a priori into a primary best-known functional pathway based on the literature (listed in Appendix A). Gene- and pathway-based *P*-values were computed using the PIGE (Self-Contained Gene Set Analysis for Gene- and Pathway-Environment Interaction Analysis) R package which implements the modified Adaptive Rank Truncated Product (ARTP) test using a permutation algorithm [33] to accommodate gene-environment interactions (https://cran.rproject.org/web/packages/PIGE/index.html). Prior to this analysis, SNPs in high linkage disequilibrium (LD) were removed using AdaJoint [34] and the online tool SNPsnap (https://data.broadinstitute.org/mpg/snpsnap/about.html) so that all SNP pairs had LD r^2^ < 0.8. Gene x Se status interactions were also examined using the PIGE R package. Although these methods do not identify individual susceptibility loci, they may help to identify a pathway that could modify the association between Se status and CRC risk. An advantage is that they do not require a priori knowledge of directionality for the variants.

All statistical tests were two-sided, and *P*-values < 0.05 were considered statistically significant (except *P* < 0.1 for *P*_ACT_). Analyses were conducted using SAS version 9.2 (SAS Institute, Cary, NC, USA) and R (R Foundation for Statistical Computing, Vienna, Austria; http://www.R-project.org/) statistical packages.

## 3. Results

### 3.1. Baseline Characteristics of Participants

The baseline characteristics of participants are presented in Table 1. Colon and rectal cancer cases were diagnosed, on average, 4.1 and 4.2 years after blood collection, respectively. CRC cases were overall less likely to be physically active compared to controls. There were no data on Se supplement use for our study participants.

### 3.2. Se Pathway Genetic Variation and Colorectal Cancer (CRC) Risk Association

The 1040 tagging SNPs successfully analyzed from 154 genes and in HWE are shown in Appendix A (which also provides all the genetic analysis results for CRC, plus stratified analyses for colon and rectal sub-site and by sex). These include 325 SNPs from 41 selenoprotein and Se transport/selenoprotein biosynthesis genes (designated as the primary Se pathway 1), and 715 variants from the other 113 wider Se metabolic pathway genes (pathways 2–8). A summary of the genetic associations before and after multiple testing corrections is provided in Appendix A. Prior to adjustment for multiple comparisons, there were 144 SNPs in 63 genes nominally associated with CRC risk (*P* < 0.05 in at least one of the disease penetrance models tested; listed in Appendix A). There were 28 unique SNPs in LD with other associated SNPs and these are listed and highlighted in Appendix A (tab ‘LD CEU’). Among the 40 SNPs significantly associated with CRC risk from pathway 1, approximately half (21) were in 12 selenoprotein genes (i.e., 50% of the 24 selenoprotein genes successfully genotyped out of 25) and have the potential to affect the function or expression of individual selenoproteins, although this remains to be investigated. These 12 selenoprotein genes include those previously found associated with CRC risk (*GPX1, GPX4, SELENOF, TXNRD1, TXNRD2, TXNRD3*; for reviews, see [4,21]) and several with limited prior or no previous evidence of association with CRC risk (*DI01, GPX6, SELENOM, SELENON, SELENOT, SELENOV*). The other 19 SNPs associated with CRC in pathway 1 are in 8 of the 17 (47%) other Se transport /selenoprotein biosynthesis genes. Therefore, they have the potential to affect the synthesis of most selenoproteins (which also needs to be examined). Notably, 31% of the genes harboring SNPs associated with CRC risk (20 of 63) were related to selenoprotein biosynthesis and function implicated in protection from cancer development [4,21] with pathway 1 and 2 proteins involved in (1) Se homeostasis (*SELENOP, SEPHS1, SEPSEC*, *EFSEC, SCLY*), (2) antioxidant enzymes (*GPXs, TXNRDs, SELENON*), and (3) endoplasmic reticulum (ER) function or stress (*SELENOF, SELENOM, SELENOT*, and again *SELENON*). Additionally, several of these genes (e.g., *GPX1, GPX5, LRP2, SEPHS1, SELENOM, SELENON, TXNRD1*, and *TXNRD2*) had multiple SNPs and/or SNPs with raw *P*-values < 0.01 associated with CRC risk further supporting a role of selenoproteins, selenoprotein metabolism, ER stress, and oxidative stress in CRC development. Table 2 lists the SNPs in the primary Se pathway 1 with raw *P*-values < 0.01 associated with CRC risk.

In pathways 3–8, considering genes with multiple SNPs associated with CRC risk or SNPs with raw *P*-values < 0.01 for at least one genetic model, there were several notable and some novel associations with CRC risk for genes in pathways 3 (*C-MYC, FRZB*), 4 (*APAF1, BAX, FOXO3*), 5 (*IL12B, RPS6KA2, TRL4*), 6 (*MSH2, MSH3*) and 7 (*BMP2, BMPR2, SMAD3, SMAD7, TFGB1*).

None of the SNPs in the primary Se pathway 1 remained significant after multiple testing corrections by the BH procedure. Overall, only 6 SNPs harbored by more distantly related genes in cell-signaling pathways retained significance (*FRZB*, *SMAD3*, and *SMAD7*; see Table 3). Genes harboring SNPs with raw *P*-values < 0.01 with CRC risk for at least one genetic model (21 genes/34 SNPs) were further considered for gene-wide variance significance by the *P*_ACT_ method. For pathway 1, the *TXNRD1* selenoprotein variant rs11111979, an intron 3′–5′UTR SNP previously associated with healthy aging [35], remained borderline significant for an association with CRC risk (*P*_ACT_ = 0.100; *P*_ACT_ significance threshold was *P* < 0.1) in the recessive genetic model. Including *FRZB*, *SMAD3* and *SMAD7*, the other wider pathway genes retaining significance were *C-MYC* (*P*_ACT_ = 0.032), *BMP2* (*P*_ACT_ = 0.012) and *BAX* (*P*_ACT_ = 0.035).

Appendix A also catalogs the SNPs with raw significant *P*-values and the BH corrections stratified by cancer sub-site (comprising 138 SNPs in 65 genes for colon in tab ‘*colon cancer*’ and 123 SNPs in 54 genes for rectum listed in tab ‘*rectal cancer*’). Additionally, the tab ‘*All*’ lists all the SNPs showing an association for CRC, colon only, rectal only, plus the analyses stratified by sex. Generally, there was predominate overlap in the genes associated with CRC and sub-site risks. Genes containing variants uniquely associated only with sub-site risk plus raw *P*-values < 0.01 comprised rs12124257 in *PTGS2* for colon cancer and 4 SNPs in *IL10* for rectal cancer.

### 3.3. Associations Between Se Pathway Genetic Variation and Se Status

Among controls, 99 different SNPs in 55 genes were nominally associated with Se status (raw *P*-values < 0.05). From these 99 variants, 87 SNPs from 45 genes were associated with either Se or SELENOP levels (55 SNPs in 33 genes and 56 SNPs in 37 genes, respectively) while the other 12 variants from 10 genes were associated with both Se status measures, including 2 each in *HIF1A* and *SMAC*. Eight pathway 1 genes harbored 14 SNPs significant for Se level changes (including 7 SNPs in 5 selenoprotein genes) while 9 pathway 1 genes carried 15 variants significant for SELENOP status association (11 SNPs in 6 selenoprotein genes). However, none of the associations retained significance after BH multiple testing adjustments. These SNP IDs together with the beta coefficients for change in Se (μg/L) or SELENOP (mg/L) are listed according to gene pathway in Appendix A.

### 3.4. Pathway Analysis

An exploratory analysis of CRC risk with gene variation and gene x Se status interaction within eight predefined pathways was performed using the PIGE package. A summary of the main PIGE results per pathway is presented in Table 4, while Appendix A provides all the *P*-values for each gene per pathway designation. Considering nominal significance for association with disease risk by pathway of *P* < 0.05, then these analyses suggest that TGF-beta signaling (*P* < 0.001) is the sole pathway highly associated with CRC risk independent of Se status interaction. Antioxidant/redox pathway genetic variation combined with Se status interactions was associated with a significant effect on CRC risk (*P* = 0.011 and 0.010 for Se and SELENOP interactions, respectively), possibly driven by SNPs in *HIF1A*, *KEAP1, GPX7, CAT*, and *SOD2* (when considering the *P*-values for each individual gene regarding gene only variation and gene x Se status interaction; see Appendix A). In contrast, the risk association with gene variation in the apoptosis pathway seems to depend more on interaction with Se levels *(P* = 0.003) but not with SELENOP concentrations (*P* = 0.105). For gene only analyses there were several genes across the pathways associated with CRC risk including the pathway 1 genes *GPX1, SELENOM, SELENON*, and *SEPHS1*, from which only *SELENON* was significant for both the gene only and gene x Se interaction PIGE analyses (Appendix A). In agreement with previous large gene association and GWAS studies, overall genetic variation in the *SMAD3, SMAD7, BMP2*, and *BMPR2* genes was associated with CRC risk [36,37,38]. Excluding individuals with no measurements of Se or SELENOP concentrations did not substantially change the main gene only PIGE results (both sets of results are provided in Appendix A).

## 4. Discussion

The results of this prospective nested case-control study represent the largest reported analysis of both the association of Se pathway SNP variation and the interaction with Se status biomarkers (serum Se levels and SELENOP protein concentrations) with CRC risk. The analysis of 1,040 tagSNPs in 154 Se pathway genes in DNA samples from 1,420 CRC cases and 1,421 controls within EPIC indicated that 144 of these SNPs in 63 genes were nominally associated with CRC risk. However, for pathway 1 only the *TXNRD1* selenoprotein gene rs11111979 SNP retained borderline significance after correction for multiple testing. For pathways 2–8, variants in *BAX, BMP2, C-MYC, FRZB, SMAD3*, and *SMAD7* passed significance thresholds following these adjustments.

Selenoprotein genes nominally associated with CRC risk included several with limited or no prior evidence (*DIO1, GPX6, SELENOM, SELENON, SELENOT, SELENOV*) and those reported in several studies (*GPX1, GPX4, SELENOF, TXNRD1, TXNRD2, TXNRD3*) for an association with CRC (or specifically colon or rectal cancer) risk. This latter group of genes has been more extensively examined due to their putative roles related to cancer prevention in colonic tissue (for reviews, see [4,21]), while the former group of selenoprotein genes have generally less well-characterized function, especially regarding how they may affect colorectal function and CRC development. Overall, any functional consequences from genetic variations in these genes, together with Se status, may affect several oxidative stress, inflammatory, and signal translation pathways implicated in colorectal carcinogenesis [13,39]. Notably several of these genes are ER-resident selenoproteins (*SELENOF, SELENOM, SELENON, SELENOT*), thought to be involved in ER-stress response and calcium flux, comprising a potentially important mechanism of selenoprotein-related cancer prevention or promotion [40].

None of the 3 *GPX1* SNPs (rs17080528, rs3448, rs9818758) or rs2074451 in *GPX4* associated with CRC risk are in high LD (i.e., r^2^ ≥ 0.8) to the functional *GPX1* Pro/Leu rs1050450 and *GPX4* rs713041 SNPs (for which the *Illumina* assays failed) previously implicated in prostate, breast, lung (rs1050450), and CRC risk (rs713041) [4]. However, from these pathway 1 genes, only a *TXNRD1* selenoprotein variant (rs11111979), one of the three thioredoxin reductases which function in redox control [8], remained borderline significant for an increased CRC risk when applying gene-wide variance considerations by the *P*_ACT_ method. Interestingly, this SNP inducing a change in the 5′untranslated region of *TXNRD1*, among others in *TXNRD1*, was previously observed to be associated with age-related physical performance [35], and age is a primary risk factor for CRC development [41]. In the wider metabolic pathway, following adjustment for multiple testing, genotypes for SNPs in Wnt, TGF-beta signaling, and apoptosis pathway genes (*C-MYC, FRZB, SMAD3, SMAD7, BMP2*, and *BAX*) were also significantly associated with CRC risk.

Positive associations of selenoprotein gene variants with CRC risk have been more commonly reported in areas with suboptimal Se availability such as European populations, than regions with generally adequate Se intake (e.g., North America). However, tagSNPs in several selenoprotein genes (*GPX3, TXNRD3, SELENON, SELENOF*, and *SELENOX*) were also associated with colon or rectal cancer risk and/or survival outcomes in two separate studies of several large case-control USA cohorts drawn from populations with generally adequate dietary Se intakes [22,23]. Associations of multiple SNPs in the same selenoprotein gene with CRC risk were observed in this study for *GPX1*, *SELENON*, *TXNRD1*, and *TXNRD2* (3, 4, 3, and 3 SNPs, respectively), broadly comparable to previous reports [4]. As reported by Slattery et al. in 2012 many of the same selenoprotein genes were separately associated with colon and rectal cancer risk in sub-site analyses although risks often differed by SNP [22]. From the 4 *SELENON* variants that were associated with CRC risk (rs11247735, rs2072749, rs4659382, and rs11247710), the first 3 were previously associated with rectal cancer risk in this North American cohort [22]. In our sub-site analyses, rs11247735, rs2072749, and rs11247710 were associated with rectal cancer risk only, and rs4659382 with both colon and rectal cancer. We also found further modifications of gene only and gene-Se risk for CRC by sex, as indicated by our previous studies of selenoprotein genetic variation in a Czech population [19] and Se status in the EPIC study [5]. This reflects the importance of interactions between Se intake, Se status and genotype, sex and CRC sub-type risks (reviewed in [4]).

Prior to this study there were few data available on the interaction of selenoprotein genotype and Se status regarding CRC risk, apart from a study in a Se replete population of North American women which reported that the null results for serum Se did not differ by selenoenzyme (*GPX1-4* and *SELENOP*) genetic variants [17]. The effect of Se pathway SNPs on the efficacy of Se utilization may be particularly relevant to CRC risk in populations with sub-optimal Se status, such as this study within EPIC [5]. We observed that numerous genetic variations were associated with Se status levels (as assessed by serum Se and SELENOP concentration), although these were not significant after adjustment for multiple tests. However, in the PIGE analysis overall gene and pathway genetic variation interacted with biomarkers of Se status to alter CRC risk. As expected, there were several variations in pathway 1 nominally associated with Se status levels. These included selenoprotein genes *SELENON*, *SELENOP*, *SELENOS*, and *TXNRD1* that are regulated by Se availability and whose genetic variations have been previously shown to affect blood and tissue Se levels [6,21,42,43]. Transgenic mouse studies underlined the critical function of SELENOP for Se organification and transport [11]. Thus, SNP interactions with SELENOP levels may be particularly important regarding CRC risk as serum SELENOP is a functional marker of Se status and is more associated with CRC risk than Se in this cohort [5]. Notably, all 4 of the variants associated with both SELENOP status and CRC risk in the SNP only analysis were from 2 selenoprotein genes; rs4659382, rs11247710, and rs2072749 in *SELENON* and the *P*_ACT_ borderline significant rs11111979 variant in *TXNRD1*. Selenoproteins SELENOP, SELENON, and TXNRD1 are antioxidant enzymes and their genetic variations plus regulation by SELENOP levels may be important factors in relation to colorectal carcinogenesis. We observed an association of SELENOP levels with rs6413428 in *SELENOP*, which in a SNP-only analysis was previously observed to be associated with CRC risk in the USA [44], an area of generally high Se status, but not in our study. Another *SELENOP* variant, rs3877899, was also associated with SELENOP status, while the GG genotype for this SNP previously showed the highest significant correlation of all selenoprotein genotypes tested between serum Se and activity of the vital antioxidant enzyme thioredoxin reductase [45]. This latter study also showed a correlation between serum Se and increased DNA damage with *SELENOS*-rs4965373 under peroxide challenge. We selected rs12910524 in *SELENOS* as a proxy tagSNP for this variant (as rs4965373 failed *Illumina* assay development) and found that it was significantly associated with Se levels. The lipoprotein megalin receptor (LRP2) protein appears to mediate SELENOP uptake to various tissues and affect plasma Se status levels [46,47]. Here, the *LRP2* SNPs rs12614394, rs2229266, rs2389557, rs700552, and rs9789747 were associated with CRC risk alone while the rs3755166 promoter SNP was associated with Se levels. Previously, rs3755166 has been associated with Alzheimer’s disease with the rare allele showing decreased transcriptional activity [48]. Intriguingly, this indicates a potential mechanism for the suggested link of sub-optimal Se status with neurodegenerative disease [49].

Supporting the data presented here, *GPX1* and *GPX4* selenoprotein gene loci have been implicated in GWAS of inflammatory bowel disease (IBD), which is a risk factor for CRC development [50,51]. Additionally, the rs7901303 variant from the selenophosphate synthetase 1 (*SEPHS1*) gene, which plays a major role in selenoprotein synthesis, was associated in this study with CRC risk (before multiple testing corrections). rs7901303 was previously associated with risk of Crohn’s disease in interaction with serum Se levels in a sub-optimal Se population of New Zealand [52]. The genetic associations identified in these studies suggest therefore a key role of the corresponding proteins in colorectal function and/or the carcinogenic process.

Genomic studies and animal models have shown Se intake to not only affect expression of selenoprotein genes but also pathways key to colorectal carcinogenesis such as the antioxidant response, immune and inflammatory pathways (including NFkB and Nrf2 signaling) and the Wnt signaling pathway [4,13,53]. Furthermore, expression of constituents of these metabolic pathways has been shown to be affected by Se level in human rectal biopsies [54]. Therefore, in addition to a focus on Se metabolism and selenoprotein genes, the present analysis also encompassed a substantial examination of genetic variations in these selenium-relevant pathways. Associations of multiple SNPs in the same gene (several of which are novel) with CRC risk were observed in genes from pathways 2–8, e.g., *BAX, GPX5, FOXO3, IL12B, TLR4, MSH2, MSH3, TGFB1*, as well as *IL10* with rectal cancer risk. Polymorphisms in several of these genes have previously been associated with CRC risk [55,56,57,58,59,60,61,62]. After BH multiple testing corrections, SNPs in cell-signaling pathways retained significance (*FRZB*, *SMAD3*, and *SMAD7*). The variants in both *SMAD* genes were previously linked to CRC by GWAS, suggesting a role of these variants in CRC development [38]. The association of rs17265803 in *FRZB* appears to be novel and it is not in LD with the functional *FRZB* genetic variant Arg324Gly (rs7775) previously reported to be associated with an increased CRC risk [63], although this was not replicated in a nested case-control study [64]. Additional genetic variants retaining significance by *P*_ACT_ were rs6983267 in *C-MYC* previously identified in a meta-GWAS [65], rs235770 in *BMP2* previously associated with colon cancer risk [37], and a novel association of rs4645887 in *BAX*.

In the pathway analysis, all the Se pathway genes were grouped into a primary best-known functional pathway and were analyzed for the association of whole gene and whole pathway genetic variation with CRC risk, and in interaction with Se status. Neither gene only variation or interactions with Se status in the core pathway 1 selenoprotein and biosynthesis pathway were associated with CRC risk by pathway, although gene only variation for *GPX1*, *SELENOM*, *SELENON*, and *SEPHS1* plus gene x Se status interactions for *SELENON* (with SELENOP) and *PSTK* (with Se) were associated with CRC risk. In the gene x Se analysis, only pathway 2 (antioxidant/redox) was significant for an association with CRC risk for both Se and SELENOP. Alternatively, it also remains possible that the genetic ‘noise’ from any irrelevant selenoproteins masked the overall risk associations for pathway 1 (based on the rationale that most genes in pathways like oxidative stress are important in cancer prevention but that some of the selenoproteins may be irrelevant to colorectal carcinogenesis, as they are included solely because they share selenocysteine motifs). This is partly supported by the strongest association (by PIGE) in pathway 1 for gene variance in *GPX1*, which has been previously implicated in risk of various cancers [4]. However, these pathway divisions cannot reflect, for example, the biological overlaps with the antioxidant selenoprotein genes in pathway 1 and their non-Se containing counterparts in pathway 2. Several aspects of our data suggest a potentially under-appreciated focus on variation in apoptosis genes (pathway 4) and CRC risk that may also be modified by Se status. These comprise the association of several SNPs in both the *FOXO3* and *BAX* genes (including the *P*_ACT_ significant rs4645887 variant in *BAX*) with CRC risk, significance of *FOXO3* for overall gene variation, and several significant findings of SNP x Se and gene x Se status interactions for genes in this pathway (e.g., *SMAC*, *CASP8*, *MAPK8*, and *MAPK9*). Overall, our analyses suggest that genetic variation in TGF beta signaling (pathway 7), which includes members such as *BMP2*, *BMPR2*, *SMAD3*, and *SMAD7* implicated in CRC risk by previous large case-control and GWAS reports [36,37,38], is sufficient to alter CRC risk, independent of Se status interaction, while SNP risk associations attributed to the antioxidant and apoptosis pathways may be significantly modified by Se status interactions.

Strengths and weaknesses of our study design for the Se status analyses have been discussed earlier [5]. The hypothesis-driven approach and appreciable sample size within a large, prospective study allowed an extensive examination of Se pathway genetic variation (including gene pathway analyses) and the interaction with robust markers of Se status regarding CRC risk. Despite the large sample size, gene pathway, gene–Se interaction analysis and some stratified analyses had limited power, particularly analyses by sex and anatomical sub-sites. The pathway designations were assigned based on known function from the literature, and there will be interactions between these pathways that we were not able to model. Finally, as most of the reported associations involve tagSNPs of no known functionality (or the actual contributing functional variant(s) they tag) additional genetic mapping and lab-based studies will be needed to explore these aspects.

## 5. Conclusions

In summary, the present study indicates that genetic variation in selenoprotein genes and genes in antioxidant/redox, Wnt, apoptotic, and TGF-beta signaling pathways may modify risk of CRC development. Furthermore, for genes in antioxidant/redox and apoptotic pathways the influence of SNPs on the disease risk is also dependent on interaction with Se status. Overall, these results taken together with our previous study [5] suggest that risk of CRC may be modified by genotype, Se status, sex, and gene variation interactions within biological pathways. Thus, will individuals harboring these genotypes benefit from increased Se intake, including consideration of ‘Se adequate’ environments, such as the US, where Se intervention trials have not shown a significant benefit in the general population [66]? Before such a recommendation can be defined, further examination of these findings in other populations and investigation of Se metabolism is needed to clarify the relevance of the Se pathway and signaling genotypes for CRC etio-pathogenesis, especially for individuals with suboptimal Se status.

## Figures and Tables

**Table 1 nutrients-11-00935-t001:** Selected baseline characteristics of incident colon and rectal cancer cases and controls, the European Prospective Investigation into Cancer and Nutrition (EPIC) study, 1992–2003.

Characteristic	Colon Cancer Cases		Rectal Cancer Cases		Controls	
N	900		520		1419	
Women, N (%)	475	(52.8)	230	(44.2)	701	(49.4)
Mean age at blood collection, (SD) yrs	58.8	(7.5)	58.0	(6.9)	58.6	(7.4)
Mean years of follow-up (SD) yrs	4.1	(2.3)	4.2	(2.2)		
Smoking status, N (%) *						
Never	385	(42.8)	195	(37.5)	594	(41.9)
Former	299	(33.2)	177	(34)	460	(32.4)
Smoker	204	(22.7)	142	(27.3)	349	(24.6)
Physical activity, N (%) *						
Inactive	129	(14.3)	73	(14)	183	(12.9)
Moderately inactive	257	(28.6)	145	(27.9)	367	(25.9)
Moderately active	374	(41.6)	209	(40.2)	612	(43.1)
Active	75	(8.3)	55	(10.6)	148	(10.4)
BMI, kg/m^2^, (SD)	26.9	(4.36)	26.6	(3.92)	26.3	(3.84)
Country, N (%)						
Sweden	55	(6.1)	33	(6.3)	86	(6.1)
Denmark	174	(19.3)	164	(31.5)	340	(24)
The Netherlands	99	(11)	54	(10.4)	158	(11.1)
United Kingdom	166	(18.4)	74	(14.2)	250	(17.6)
Germany	110	(12.2)	69	(13.3)	169	(11.9)
France	22	(2.4)	6	(1.2)	29	(2)
Italy	144	(16)	58	(11.2)	198	(14)
Spain	101	(11.2)	45	(8.7)	141	(9.9)
Greece	29	(3.2)	17	(3.3)	48	(3.4)

* Percentages do not add up to 100% due to missing values. Abbreviations: BMI, body mass index; N, sample size; SD, standard deviation; yrs, years.

**Table 2 nutrients-11-00935-t002:** Single nucleotide polymorphisms (SNPs) associated with colorectal cancer (CRC) risk in primary selenium pathway 1 (selenium and selenoprotein transport, biosynthesis and metabolism) with raw *P*-values < 0.01 in at least one genetic model prior to multiple testing adjustment, the EPIC study, 1992–2003.

Gene/SNP/Genotype	CRC	Control	OR (95% CI)	*P*	*P_BH_^+^*
*GPX1*/rs17080528					
GG	700	620	1.00 (ref)	0.010	0.703
GA	580	636	0.81 (0.69,0.95)		
AA	131	154	0.75 (0.58,0.97)		
Additive *	1411	1410	0.84 (0.75,0.95)	0.003	0.554
Dominant (GA + AA vs. GG)	1411	1410	0.80 (0.69,0.92)	0.003	0.534
Recessive (AA vs. GG + GA)	1411	1410	0.83 (0.65,1.06)	0.137	0.854
*SELENOM*/rs11705137					
AA	367	346	1.00 (ref)	0.024	0.753
AG	631	648	0.91 (0.75,1.09)		
GG	288	359	0.74 (0.60,0.92)		
Additive *	1286	1353	0.86 (0.78,0.96)	0.008	0.684
Dominant (AG + GG vs. AA)	1286	1353	0.85 (0.71,1.01)	0.064	0.815
Recessive (GG vs. AA + AG)	1286	1353	0.79 (0.66,0.95)	0.012	0.710
*SELENON*/rs4659382					
GG	783	713	1.00 (ref)	0.019	0.747
GC	509	573	0.80 (0.69,0.94)		
CC	96	107	0.82 (0.61,1.10)		
Additive *	1388	1393	0.86 (0.76,0.97)	0.011	0.710
Dominant (GC + CC vs. GG)	1388	1393	0.81 (0.69,0.94)	0.005	0.625
Recessive (CC vs. GG + GC)	1388	1393	0.90 (0.67,1.20)	0.455	0.954
*SEPHS1*/rs2275129					
GG	361	423	1.00 (ref)	0.032	0.780
GC	726	690	1.23 (1.03,1.47)		
CC	321	295	1.28 (1.04,1.58)		
Additive *	1408	1408	1.14 (1.02,1.26)	0.017	0.747
Dominant (GC + CC vs. GG)	1408	1408	1.25 (1.05,1.47)	0.010	0.697
Recessive (CC vs. GG + GC)	1408	1408	1.12 (0.94,1.34)	0.217	0.885
*TXNRD1*/rs11111979 ^^^					
GG	395	429	1.00 (ref)	0.015	0.745
GC	627	680	1.00 (0.84,1.20)		
CC	279	230	1.34 (1.07,1.67)		
Additive *	1301	1339	1.14 (1.02,1.27)	0.022	0.749
Dominant (GC + CC vs. GG)	1301	1339	1.09 (0.92,1.28)	0.315	0.932
Recessive (CC vs. GG + GC)	1301	1339	1.33 (1.10,1.62)	0.004	0.566

^+^ = After Benjamini–Hochberg (BH) multiple testing correction; * = Additive models impose a structure in which each additional copy of the variant allele increases the response (log odds ratio) by the same amount; ^ = *TXNRD1* rs11111979 was borderline significant after adjustment for correlated tests (*P_ACT_* = 0.10). EPIC = European Prospective Investigation into Cancer and Nutrition.

**Table 3 nutrients-11-00935-t003:** Single Nucleotide Polymorphisms (SNPs) statistically significantly associated with colorectal cancer (CRC) risk after Benjamini–Hochberg (BH) multiple testing correction, the EPIC study, 1992–2003.

Gene/SNP/Genotype	CRC	Control	OR (95% CI)	*P*	*P_BH_*
*FRZB*/rs17265803 ^					
AA	844	976	1.00 (ref)	3.04E-06	0.003
AG	315	240	1.56 (1.28,1.89)		
Additive *	1163	1232	1.35 (1.13,1.62)	0.001	0.372
Dominant (AG + GG vs. AA)	1163	1232	1.48 (1.22,1.79)	6.77E-05	0.034
Recessive (GG vs. AA + AG)	1163	1232	0.26 (0.09,0.80)	0.018	0.747
*SMAD3*/rs7180244 ^					
GG	994	1183	1.00 (ref)	2.22E-16	1.12E-12
GC	372	198	2.33 (1.92,2.83)		
Additive *	1372	1388	2.16 (1.79,2.60)	1.11E-15	3.74E-12
Dominant (GC + CC vs. GG)	1372	1388	2.28 (1.88,2.77)	1.11E-16	1.12E-12
Recessive (CC vs. GG + GC)	1372	1388	0.87 (0.29,2.61)	0.804	0.997
*SMAD7*/rs11874392					
AA	478	400	1.00 (ref)	1.39E-07	2.82E-04
AT	704	671	0.88 (0.74,1.04)		
TT	222	337	0.55 (0.44,0.68)		
Additive *	1404	1408	0.75 (0.68,0.84)	1.99E-07	3.36E-04
Dominant (AT + TT vs. AA)	1404	1408	0.77 (0.65,0.90)	0.001	0.372
Recessive (TT vs. AA + AT)	1404	1408	0.59 (0.49,0.72)	6.26E-08	1.58E-04
*SMAD7*/rs12953717					
GG	366	470	1.00 (ref)	3.47E-05	0.019
GA	671	643	1.36 (1.14,1.62)		
AA	335	273	1.60 (1.29,1.98)		
Additive *	1372	1386	1.27 (1.14,1.41)	9.07E-06	0.006
Dominant (GA + AA vs. GG)	1372	1386	1.43 (1.21,1.68)	2.38E-05	0.014
Recessive (AA vs. GG + GA)	1372	1386	1.32 (1.10,1.59)	0.003	0.534
*SMAD7*/rs4939827					
AA	433	378	1.00 (ref)	6.46E-06	0.005
AG	664	634	0.92 (0.77,1.09)		
GG	248	357	0.60 (0.49,0.75)		
Additive *	1345	1369	0.79 (0.71,0.87)	9.46E-06	0.006
Dominant (AG + GG vs. AA)	1345	1369	0.80 (0.68,0.95)	0.009	0.697
Recessive (GG vs. AA + AG)	1345	1369	0.64 (0.53,0.77)	1.66E-06	0.002
*SMAD7*/rs6507874					
AA	467	389	1.00 (ref)	1.93E-06	0.002
AG	705	677	0.87 (0.73,1.03)		
GG	234	336	0.57 (0.46,0.71)		
Additive *	1406	1402	0.77 (0.69,0.86)	1.26E-06	0.002
Dominant (AG + GG vs. AA)	1406	1402	0.77 (0.65,0.90)	0.001	0.420
Recessive (GG vs. AA + AG)	1406	1402	0.63 (0.52,0.76)	1.16E-06	0.002

^ = Results for the rare homozygous genotypes are omitted for these SNPs due to the small sample numbers with these genotypes; * = Additive models impose a structure in which each additional copy of the variant allele increases the response (log odds ratio) by the same amount. EPIC = European Prospective Investigation into Cancer and Nutrition.

**Table 4 nutrients-11-00935-t004:** *P*-values for genetic pathways and pathway-selenium (Se) status interactions and colorectal cancer risk, the EPIC study, 1992–2003.

Pathway	*P* _Pathway Only_	*P* _Pathway Only (non-Missing Se Status)_ ^***^	*P* _Pathway x Se Interaction_	*P* _Pathway x SELENOP Interaction_
Se and Selenoproteins *	0.217	0.098	0.615	0.726
Antioxidant and Redox	0.173	0.072	0.011	0.010
Cell signaling **	0.307	0.489	0.223	0.872
Apoptosis	0.361	0.097	0.003	0.105
Inflammation	0.822	0.262	0.199	0.607
DNA repair	0.739	0.432	0.175	0.088
TGFβ signaling	<0.001	0.001	0.061	0.764
Cell cycle control	0.398	0.475	0.097	0.449

* Se and selenoprotein transport, biosynthesis & metabolism. ** Includes Wnt, mTOR, NfkB, and Nrf2 signaling. *** Includes only participants with non-missing blood Se or SELENOP concentrations. EPIC = European Prospective Investigation into Cancer and Nutrition.

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
