# Peer review of "Association of Selenoprotein and Selenium Pathway Genotypes with Risk of Colorectal Cancer and Interaction with Selenium Status"

_nutrients, 2019, doi:10.3390/nu11040935_

Reviewer 1 Report

This is a very nicely written paper that explores associations between selenoprotein and selenium pathway genotypes and risk of colorectal cancer.  A major strength of the manuscript is that interactions with selenium status and selenoprotein P levels are also examined.

Minor comments:

1) Please expand on whether the study population resided in a high or low selenium status area.  It would be nice to mention the population selenium and selenoprotein P levels, even if reported elsewhere.  Are these values high or low?

2) Do you have information on whether subjects took selenium supplements?  

3) Am I correct, that the study population was not followed by any special screening procedure, such as colonoscopy?

4) The genetic analyses and methods are first rate.

5) In the supplemental figures, I don't understand the red and yellow color codes.  Please explain.

6) The page numbering goes from 1 to 10, then starts over again at 1.

7) In the PIGE analysis, several selenoproteins including SELENON, SELNOP and SELENOP were nominally associated with selenium status.  Are these perhaps linked by a role in endoplasmic reticulum stress, a pathway that seems to be particularly important for selenoproteins and is only touched on in the discussion and pathway classifications?

8) There is a formatting problem with reference 5.

Author Response

We wish to warmly thank the reviewers for their very supportive and insightful comments. It is a welcome boost to have such enthusiasm and interest in our work shown by our colleagues like this. It is our pleasure to respond to these comments and insights.

Reviewer 1

This is a very nicely written paper that explores associations between selenoprotein and selenium pathway genotypes and risk of colorectal cancer.  A major strength of the manuscript is that interactions with selenium status and selenoprotein P levels are also examined.

Minor comments:

1) Please expand on whether the study population resided in a high or low selenium status area.  It would be nice to mention the population selenium and selenoprotein P levels, even if reported elsewhere.  Are these values high or low?

Response: We thank the reviewer for highlighting this important issue. The reviewer is correct to point out the need to mention if the study population resided in a high or low selenium status area.

As highlighted in the introduction, baseline selenium levels in a European population are generally low/suboptimal, and in this study population we have previously described that the selenium and selenoprotein P levels were suboptimal and associated with CRC risk (Hughes et al, 2015; ref 5 in manuscript). We have now added in the Introduction (lines 136-140 and lines 159-168) that our study was within a generally suboptimal population and provided the mean selenium and selenoprotein P levels in the assessed cases and controls.

In a population with insufficient Se supply there is a strong and linear relation between the Se and SELENOP concentrations, which we observe in our study population (discussed in Hughes et al, 2015). This was as similarly shown in the British study of Hurst and Fairweather-Tait (et al) (Establishing optimal selenium status: results of a randomized, double-blind, placebo-controlled trial. Am J Clin Nutr. 2010;91(4):923-31) where supplemental Se increased both serum Se and SELENOP concentrations. This Nutrients article also clearly highlights that Europe is not sufficiently supplied with selenium; Stoffaneller R, Morse NL. A Review of Dietary Selenium Intake and Selenium Status in Europe and the Middle East. Nutrients;7(3):1494–1537.

Lines 136-140: ‘Data from nutritional intervention trials and epidemiological studies suggest implications for Se intake regarding CRC risk could potentially be more important in individuals with particular selenoprotein genotypes and/or in populations with low Se status, such as in Western Europe where the present study was conducted [4,5,17,18].’

Lines 159-168: ‘In this study, we have examined for the first time the association of detailed Se pathway gene variation with cancer risk in 1,420 CRC cases and 1,421 controls within the European Prospective Investigation into Cancer and Nutrition (EPIC) cohort. We previously reported in a subset of this nested cohort with 966 case-control pairs that a higher Se status (ascertained by serum levels of Se and SELENOP) was associated with a lower CRC risk [5]. In these Western European subjects, the mean Se and SELENOP circulating levels were 84.0 μg/L and 4.3 mg/L in cases and 85.6 μg/L and 4.4 mg/L in controls, respectively. Thus, our present study was conducted in a generally suboptimal Se status population, as these Se concentrations are insufficient for optimal GPX3 activity and SELENOP saturation [5,6]. We now report the interaction between these genes and their corresponding pathways with Se status biomarkers and CRC risk.’

2) Do you have information on whether subjects took selenium supplements? 

Response: We agree with the reviewer that this would be very useful information, unfortunately, we do not have data on supplement use for the subjects in our study. We have clarified this point in line 327 of the results section. ‘There were no data on Se supplement use for our study participants’.

Thus, the measured circulating levels of Se and SELENOP will reflect a total exposure to Se including diet and any supplement use.

However, there is data on Se supplement use in sub cohorts within EPIC (second follow up), which is in the range of 4.3 (vegetarians) to 7.4% (vegans) [4.8% for ‘meat eaters’ and 5.6% for ‘fish eaters’). Please compare: table 5 in “High compliance with dietary recommendations in a cohort of meat eaters, fish eaters, vegetarians, and vegans: results from the European Prospective Investigation into Cancer and Nutrition–Oxford study (Sobiecki JG, Appleby PN, Bradbury KE, Key TJ. Nutr Res. 2016;36(5):464-77).

3) Am I correct, that the study population was not followed by any special screening procedure, such as colonoscopy?

Response: Yes, the reviewer is correct that the study population was not followed by any special screening procedure, such as colonoscopy. This would again be useful data to have (to assess association or confounding with adenoma presence for example) but we only have follow-up data for cancer diagnosis.

4) The genetic analyses and methods are first rate.

Response: We warmly thank the reviewer for this very supportive comment.

5) In the supplemental figures, I don't understand the red and yellow color codes.  Please explain.

Response: We are glad to clarify this and thank the reviewer for pointing out the red color code (this was our mistake, these should all be yellow color coded). This is explained in the note for the SNP tab in the table legend (‘SNPs in high linkage disequilibrium (R2 > 0.8)with other SNPs in the same gene are highlighted in yellow and listed on a separate tab with their proxy SNPs’).

6) The page numbering goes from 1 to 10, then starts over again at 1.

Response: Our apologies and we thank the reviewer for noticing this, which happened in the manuscript word template for the journal after having to change the pages for table 2 to a landscape orientation. Now we have simplified this table (it is now new table 3) it is in the portrait format and this error has been corrected.

7) In the PIGE analysis, several selenoproteins including SELENON, SELNOP and SELENOP were nominally associated with selenium status.  Are these perhaps linked by a role in endoplasmic reticulum stress, a pathway that seems to be particularly important for selenoproteins and is only touched on in the discussion and pathway classifications?

Response: We thank the reviewer for this insightful hypothesis, which we would also consider to be a very possible mechanism involving these selenoproteins in cellular protection (that these proteins may play a crucial role in the ER stress response). However, only SELENON was nominally associated with selenium status (SELENOP) in the PIGE analysis (Table S5) but the P value was only 0.043. Overall genetic variation in SELENOM, SELENON, and SELENOT (relevant to the ER) where associated in the PIGE analysis with CRC risk (in at least one of the SNP only analysis models). And individual SNPs in these three genes (including 4 SNPs in SELENON) and in SELENOF were nominally associated with CRC risk (P<0.05) before correction for multiple testing, but not afterwards. So, there may well be something in respect of ER function but the picture is not clear and we didn’t want to over-emphasize these results and over-speculate on them.

However, we do mention in section 3.2 of the results (in lines 354-358) that ‘Notably, 31% of the genes harboring SNPs associated with CRC risk (20 of 63) were related to selenoprotein biosynthesis and function implicated in protection from cancer development [4,21] with pathway 1 and 2 proteins involved in (1) Se homeostasis (SELENOP, SEPHS1, SEPSEC, EFSEC, SCLY), (2) antioxidant enzymes (GPXs, TXNRDs, SELENON), and (3) endoplasmic reticulum (ER) function or stress (SELENOF, SELENOM, SELENOT, and again SELENON).

We have now also added ER stress to our summary sentence in section 3.2 (lines 359-363):

Additionally, several of these genes (e.g., GPX1, GPX5, LRP2, SEPHS1, SELENOM, SELENON, TXNRD1, and TXNRD2) had multiple SNPs and/or SNPs with raw P-values<0.01 associated with CRC risk further supporting a role of selenoproteins, selenoprotein metabolism, ER stress, and oxidative stress in CRC development’.

We have also now added in the discussion section new sentences summarising selenoprotein function relevant to colorectal carcinogenesis including this aspect for the ER (lines 465-470). We have also included here two additional reference, one to  a review of selenoproteins in colon cancer [ref 39] and the second [ref 40] to some very interesting work from Matthew Pitts and Peter Hoffmann* that suggests ER-resident selenoproteins are regulators of calcium signaling and homeostasis; such mechanisms may form a part of the cancer preventative and also promotive (for cancer progression) differential roles of selenoproteins such as SELENOK and SELENOF.

lines 465-470: ‘Overall, any functional consequences from genetic variations in these genes, together with Se status, may affect several oxidative stress, inflammatory, and signal translation pathways implicated in colorectal carcinogenesis [13, 39]. Notably several of these genes are ER-resident selenoproteins (SELENOF, SELENOM, SELENON, SELENOT), thought to be involved in ER-stress response and calcium flux, comprising a potentially important mechanism of selenoprotein-related cancer prevention or promotion [40]’.

39. Peters, K.M.; Carlson, B.A.; Gladyshev, V.N.; Tsuji, P.A. Selenoproteins in colon cancer. Free Radic. Biol. Med. 2018;127:14-25.

40. Pitts, M.W.; Hoffmann, P.R. Endoplasmic reticulum-resident selenoproteins as regulators of calcium signaling and homeostasis. Cell calcium 2018;70:76-86.

8) There is a formatting problem with reference 5.

Response: We don’t see any evident problem with the format of reference 5.

Reviewer 2 Report

Colorectal cancer (CRCs) displays a major health burden and metastasis and disease recurrence remain challenging. Risk factors underlying the development of this disease include (epi-)genetic alterations, inflammatory bowel diseases as well as diet and life style. In fact, suboptimal dietary intake rates of selenium (Se) were associated with CRC risk. In their well-written manuscript, the authors aimed to elucidate the correlation of gene variations in Se-associated genes with the risk of CRC development. They have used a comprehensive approach by screening a large patient cohort for single nucleotide polymorphisms (SNPs) in genes encoding selenoproteins and Se pathway genes. Their findings on this topic indicate the relevance of Se and associated genes in the development of CRC and will be of interest to the readership of Nutrients. A number of minor points should be addressed prior to publication as detailed below.

1.      The authors start the results section by introducing the patient cohort and the differences between the healthy controls and the colon/rectal cancer patients. They conclude that the BMI of diseased individuals (26.9 or 26.6) is slightly higher than the BMI of healthy individuals (26.3). Since the difference between those cohorts is extremely small and is very unlikely to have any biological consequences, it is not worth mentioning.

2.      In section 3.2 the authors demonstrate which SNPs and genes were associated with CRC risk and which pathways they belong to. This paragraphs includes many numbers, percentages and gene names and could be unclear to the reader. Since only a relatively small number of genes is discussed in this section, a table showing these genes and the corresponding pathways/functions in the main manuscript would give a good overview and will make these results more clear.

3.      The pathway 4 member “FOX03” was mentioned, which should probably be “FOXO3” instead.

4.      Please define meanings of acronyms the first time they are being used.

Author Response

We wish to warmly thank the reviewers for their very supportive and insightful comments. It is a welcome boost to have such enthusiasm and interest in our work shown by our colleagues like this. It is our pleasure to respond to these comments and insights.

Reviewer 2:

Colorectal cancer (CRCs) displays a major health burden and metastasis and disease recurrence remain challenging. Risk factors underlying the development of this disease include (epi-)genetic alterations, inflammatory bowel diseases as well as diet and life style. In fact, suboptimal dietary intake rates of selenium (Se) were associated with CRC risk. In their well-written manuscript, the authors aimed to elucidate the correlation of gene variations in Se-associated genes with the risk of CRC development. They have used a comprehensive approach by screening a large patient cohort for single nucleotide polymorphisms (SNPs) in genes encoding selenoproteins and Se pathway genes. Their findings on this topic indicate the relevance of Se and associated genes in the development of CRC and will be of interest to the readership of Nutrients. A number of minor points should be addressed prior to publication as detailed below.

1.      The authors start the results section by introducing the patient cohort and the differences between the healthy controls and the colon/rectal cancer patients. They conclude that the BMI of diseased individuals (26.9 or 26.6) is slightly higher than the BMI of healthy individuals (26.3). Since the difference between those cohorts is extremely small and is very unlikely to have any biological consequences, it is not worth mentioning.

Response: We fully agree with the reviewer’s comment that these BMI differences are unlikely to be of biological significance. We have thus now deleted this sentence.

2.      In section 3.2 the authors demonstrate which SNPs and genes were associated with CRC risk and which pathways they belong to. This paragraph includes many numbers, percentages and gene names and could be unclear to the reader. Since only a relatively small number of genes is discussed in this section, a table showing these genes and the corresponding pathways/functions in the main manuscript would give a good overview and will make these results clearer.

Response: We thank the reviewer for this very helpful suggestion and agree it would make assessment of these data clearer. We did try to do this by the inclusion of supplementary figure 1 (a summary of the genetic associations before and after multiple testing corrections is provided in Supplementary Figure S1). However, we agree a table in the main text would also be helpful, so we have now added this table (new table 2) to include the genes discussed in this section for the core selenium pathway 1 (Se and selenoprotein transport, biosynthesis & metabolism, and oxidative response). To keep the table to a more manageable and readable size, and for having a more stringent P-value cut-off, we have only included the SNPs for this primary selenoprotein pathway that had raw P-values<0.01.< span="">

Thus, the former tables 2 and 3 are now tables 3 and 4. Also, to simplify the tables and keep consistency with the new table 2, we have deleted the columns for colon cancer and rectal cancer that we had in the old table 2 (new table 3). These data can be found in the supplementary files as indicated in the text.

Table 2: SNPs associated with colorectal cancer (CRC) risk in primary selenium pathway 1 (selenium and selenoprotein transport, biosynthesis and metabolism) with raw P-values<0.01 in at least one genetic model prior to multiple testing adjustment, the EPIC study, 1992-2003.

3.      The pathway 4 member “FOX03” was mentioned, which should probably be “FOXO3” instead.

Response: We thank the reviewer for their keen eye in spotting this error, and we are glad to have now been able to correct this.

4.      Please define meanings of acronyms the first time they are being used.

Response: We thank the reviewer for noticing abbreviations which we didn’t explain at first usage. Our apologies. We have now gone through the manuscript to correct this, as listed below. There were several gene names listed with their standard abbreviated nomenclature. However, we feel it would be unwieldy / messy to write all these gene names out in full followed by their abbreviations.

Single nucleotide polymorphisms (SNPs) – in the abstract (the abbreviation was provided at first usage in the introduction).

International Agency for Research on Cancer (IARC, Lyon, France)

PIGE (Self Contained Gene Set Analysis for Gene- And Pathway-Environment Interaction Analysis)

Wingless/Integrated (Wnt) and Transforming growth factor (TGF)-beta signaling